# Structural Equation Modeling of the Marine Ecological System in Nanwan Bay Using SPSS Amos

Jung-Fu Huang [1,*], Chen-Tung Arthur Chen [2], Meng-Hsien Chen [2], Shih-Lun Huang [3] and Pi-Yu Hsu [1]

1   Department and Graduate Institute of Aquaculture, National Kaohsiung University of Science and Technology, Kaohsiung 811213, Taiwan; byhsu@nkust.edu.tw
2   Department of Oceanography, National Sun Yat-sen University, Kaohsiung 804201, Taiwan; ctchen@mail.nsysu.edu.tw (C.-T.A.C.); mhchen@mail.nsysu.edu.tw (M.-H.C.)
3   Center for Data Science, New York University, New York, NY 10011, USA; sh7008@nyu.edu
*   Correspondence: jfhuang@nkust.edu.tw; Tel.: +886-07-3617141 (ext. 23703)

**Abstract:** To ensure the sustainability of the marine environment, it is crucial to understand the intricate relationship between environmental factors and marine biota. Human activities have been recognized as significant contributors to profound changes in marine ecology. However, these observable alterations often represent a cumulative effect that intertwines with less apparent natural influences. This research delved into the relationships between environmental factors and marine life in the waters adjacent to Nanwan Bay, Kenting, Taiwan. Specifically, it examined the linear relationships and the degree of changes between environmental factors and marine life. To achieve these objectives, factor analysis was employed to uncover potential latent variables that could impact marine organisms, with these variables named based on previous studies and related literature. The findings led to the development of a structural equation model (SEM) to represent the marine ecology of Nanwan Bay. The results accentuated the significant influence of primary productivity and nutrient levels on the assemblage of marine life. The application of SEM methodology sheds more light on the degree of impact natural and anthropogenic interference have on marine ecosystems.

**Keywords:** factor analysis; structural equation model; marine ecology

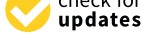


## 1. Introduction

In marine ecosystems, changes in the water temperature reflect the heat balance of seawater and its temporal variation and are also important factors affecting the survival of marine organisms [1]. To support national economic development and industrial growth, many countries, including the United States, China, and Japan, have chosen to build nuclear power plants in coastal areas to supply electricity for daily life and industry. The large amount of heated water discharged from nuclear power plants elevates the sea water temperature, potentially affecting marine organisms. However, the extent of this impact on the local region can also be influenced by site characteristics and environmental background [1]. Previous studies have shown that environmental factors can affect the abundance of marine organisms [2,3]. For instance, the water temperature, light, and nutrient concentration often affect the aggregation structure of phytoplankton, the primary producer in marine ecosystems, in specific times and spaces [4]. Hence, the increase in water temperature caused by discharged heated water may affect the photosynthetic efficiency and species composition of phytoplankton, and may also affect the metabolism of zooplankton, reducing their abundance [5–7]. Moreover, changes in prey species and water temperature may cause differences in the distribution of fish [8].

This study was conducted in the semi-enclosed Nanwan Bay, Taiwan, where about ten million tons of cooling seawater have been drawn and discharged every day since May 1984 for the operation of a nuclear power plant. Notably, Kenting National Park was established in 1982, shortly after construction began on the power plant in 1981. The operation of the

nuclear plant has since raised significant environmental concerns, leading to active research in the area. Long-term water quality monitoring data indicate that the water temperature changes in the Bay are mainly affected by weather, seasonality, and large-scale ocean events, and are not directly related to the discharge of heated water [9,10]. In an attempt to gain a deeper understanding of how these complex natural phenomena can impact marine ecosystems, it is necessary to investigate the unique influence of strongly interrelated factors. Consequently, this research aims to ascertain the extent of influences wielded by these elusive environmental factors, specifically nutrients and upwelling current, on marine life.

## 2. Materials and Methods

In this study, we analyzed a total of 223 items of valid water quality and marine biological data surveyed at 4 stations in Nanwan Bay, Kenting, Taiwan, in the same seasons and months, specifically February, May, August, and November from 2000 to 2016 (Figure 1). We employed a multi-stage approach to examine the interrelationships among various factors in the marine environment, with the aim of constructing a structural equation model (SEM), a model which integrates factor analysis and path analysis to analyze data [11,12]. This methodology can not only serve as an approach for theoretical verification, but also incorporates multiple environmental and biological factors into one model. In the first stage, Pearson correlation analysis was used to evaluate the linear association between the variables. Then, factor analysis was conducted to identify common factors among the variables, with these factors subsequently serving as measurement models in SEM.

One issue that can compromise the validity of factor analysis is excessively high or low correlations between variables. To address this, the Kaiser–Meyer–Olkin (KMO) measure, a statistic that compares the magnitude of observed correlation coefficients to the magnitude of partial correlation coefficients, was utilized [13]. In conjunction, Bartlett's test of sphericity, which checks the overall significance of all the correlations within the correlation matrix, was used. The tests helped to ensure adequate sampling (Table 1) and sufficient correlation matrices, respectively.

**Table 1.** KMO measurement sampling adequacy criteria (Kaiser, 1974) [12].

| KMO Value | Applicability of Factor Analysis |
|---|---|
| $0.9 \leq$ KMO | Marvelous |
| $0.8 \leq$ KMO $\leq 0.9$ | Meritorious |
| $0.7 \leq$ KMO $\leq 0.8$ | Middling |
| $0.6 \leq$ KMO $\leq 0.7$ | Mediocre |
| $0.5 \leq$ KMO $\leq 0.6$ | Miserable |
| KMO $\leq 0.5$ | Unacceptable |

To ensure the validity of the measurement models, we assessed the construct validity. Construct validity encompasses convergent validity, which confirms whether or not measures that should theoretically be related are indeed correlated with corresponding factors, and discriminant validity, which measures how much a construct is distinct from other constructs. We adopted factor loadings of 0.5 and above as a benchmark for convergent validity and specified cross-loadings of below 0.5 to ensure discriminant validity, as suggested by Chen (2005) [14]. Furthermore, we employed communality to quantify the extent to which a variable contributes to a latent factor. The value ranges from 0 to 1, with higher values indicating that the variable is more closely related to the common factor and has lower uniqueness. Thus, variables with high communality are deemed to be more appropriate measurement variables. Consequently, variables with a communality score exceeding 0.5 can be identified as ideal measurement variables, following Chen's

(2005) [14] guideline. After validating these measurements, environment-related literature and ecological knowledge were incorporated to assign meanings to each extracted factor.

| GPS of sampling stations of water quality and plankton | | | GPS of sampling stations of fish | | |
|---|---|---|---|---|---|
| No. | N | E | No. | N | E |
| 22 | 21°57′18″ | 120°45′44″ | 1 | 21°56′37″ | 120°45′00″ |
| 23 | 21°56′45″ | 120°45′29″ | 2 | 21°56′18″ | 120°44′47″ |
| 24 | 21°55′47″ | 120°44′56″ | 3 | 21°55′37″ | 120°44′37″ |
| 12 | 21°54′53″ | 120°44′57″ | 5 | 21°55′17″ | 120°44′27″ |

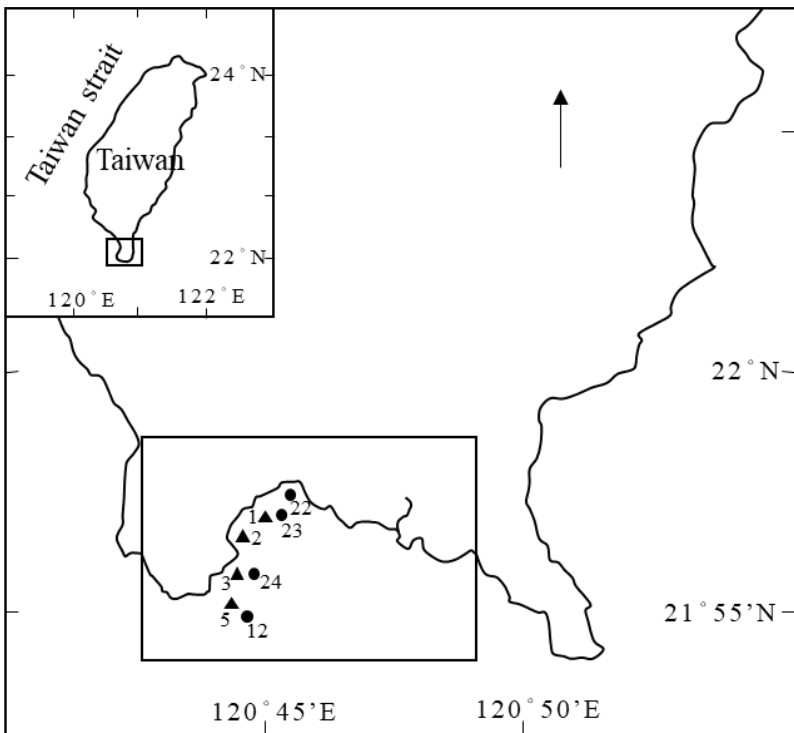

**Figure 1.** The location of research geography and sampling stations. ● Sampling station of water quality and plankton; ▲ Sampling station of fish.

Finally, potential SEMs were created based on previous studies and the results of our factor analysis (Figure 2). Relationships between marine ecological factors were determined using analytical findings from Ramdani et al. (2009) [4]. Each factor was assigned a composite score, representing the intensity of the variables it represented.

We assessed the performance of the model using absolute fit indices, incremental fit indices, and parsimonious fit indices. The absolute fit indices, such as Chi-square index and goodness of fit index, measure the degree to which the observed covariance or correlation matrix matches the predicted theoretical model. Incremental fit indices, such as the non-normed fit index, compare the fit of the theoretical model to a baseline or null model. Parsimonious fit indices, such as the parsimonious normed fit index and Hoelter's critical N, favor models that achieve a good fit with fewer parameters. Table 2 provides a summary of the model's adequacy evaluation.

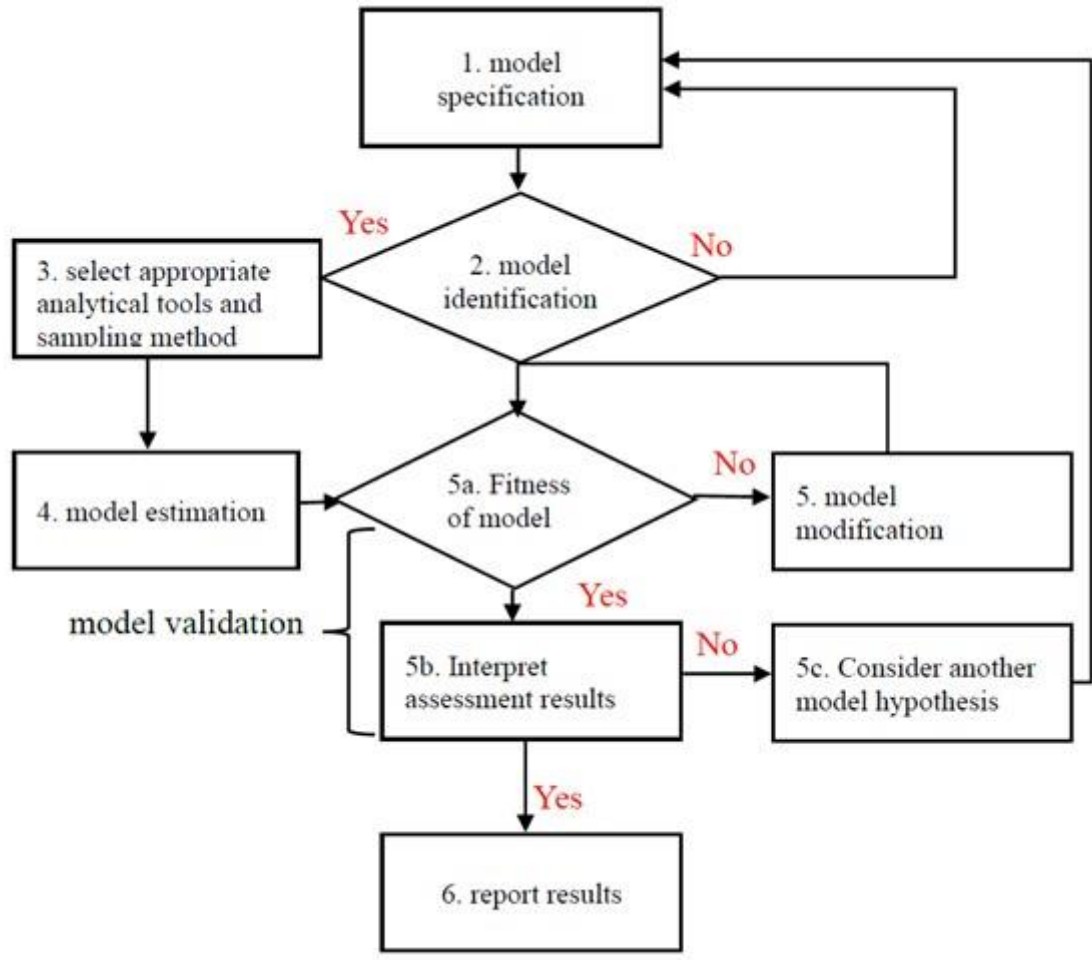

**Figure 2.** The construction process of SEM.

**Table 2.** The indices' thresholds for SEM evaluation.

| Key Metrics | | | Guidelines | Reference |
|---|---|---|---|---|
| Preliminary Fit Criteria | | Factor Loading | 0.50~0.95 | Chen, 2005 [14] |
| Overall Model Fit | Absolute Fit Indices | $\chi^2$ | The lower the better | Hwang, 2004,2009 [15,16] |
| | | $\chi^2/df$ | <5 (<3 better fit) | Hair et al., 1998 [17] Carmines et al., 1981 [18] |
| | | GFI | >0.9 | Joreskog & Sorbom, 1996 [19] |
| | | AGFI | >0.9 (>0.8 acceptable fit) | Hair et al., 1998 [17] Joreskog & Sorbom, 1996 [19] |
| | | RMSEA | <0.05, good fit | Hair et al., 2006 [20] Browne & Cudeck, 1993 [21] |
| | | | 0.05~0.08, reasonable fit | |
| | | | 0.08~0.10, medium fit | |
| | | | >0.10, poor fit | |
| | | SRMR | ≤0.08 | Hu & Bentler, 1999 [22] |
| | Incremental Fit Indices | NNFI | >0.9 | Tucker & Lewis, 1973 [23] |
| | Parsimonious Fit Indices | PNFI | ≥0.5 | Tucker & Lewis, 1973 [24] |
| | | PGFI | ≥0.5 | Mulaik et al., 1989 [25] |
| | | CN | ≥200 | Mulaik et al., 1989 [26] |

## 3. Results and Discussions

### 3.1. Pearson Correlation Coefficient

The statistics of each measurement variable are shown in Tables 3 and 4. The initial investigation relied on simple correlation analysis (Table 5) due to the variance in measurement units between water quality parameters and biometric parameters. The standardized correlation coefficient (r) was utilized to evaluate the level of linear correlation between each measurement variable and was employed as a benchmark for the development and refinement of subsequent models.

**Table 3.** Statistical summary of environmental variables.

| Environment Variable N = 223 | Unit | Minimum | Maximum | Mean | Standard Deviation | Variance |
|---|---|---|---|---|---|---|
| Temperature | °C | 17.200 | 31.100 | 26.744 | 2.078 | 4.316 |
| Salinity | psu | 32.111 | 35.741 | 34.114 | 0.498 | 0.248 |
| pH | - | 7.953 | 8.195 | 8.077 | 0.036 | 0.001 |
| Dissolved oxygen | mg/L | 5.168 | 7.560 | 6.521 | 0.281 | 0.079 |
| Transparency | m | 0.000 | 20.000 | 11.486 | 3.129 | 9.792 |
| Chlorophyll a | μg/L | 0.005 | 1.223 | 0.212 | 0.200 | 0.040 |
| Nitrate | μM | 0.000 | 2.957 | 0.616 | 0.518 | 0.268 |
| Nitrite | μM | 0.000 | 0.244 | 0.066 | 0.041 | 0.002 |
| Phosphate | μM | 0.006 | 0.780 | 0.093 | 0.067 | 0.005 |
| Silicate | μM | 0.913 | 5.630 | 2.258 | 0.756 | 0.571 |

**Table 4.** Statistical summary of biological variables.

| Biological Variable N = 223 | Unit | Minimum | Maximum | Mean | Standard Deviation | Variance |
|---|---|---|---|---|---|---|
| Fish species | Species | 15 | 67 | 39.776 | 8.699 | 75.670 |
| Fish abundance | ind./station | 51 | 3082 | 299.170 | 275.870 | 76,104.016 |
| Zooplankton | ind./1000 m$^3$ | 22,590 | 2,329,724 | 410,051.466 | 380,477.250 | $1.45 \times 10^{11}$ |
| Phytoplankton | ind./1000 m$^3$ | 10 | 120,600 | 2390.063 | 13,069.235 | $1.71 \times 10^8$ |
| Crab larvae | ind./1000 m$^3$ | 0 | 111,962 | 2496.565 | 7862.203 | $6.18 \times 10^7$ |
| Shrimp larvae | ind./1000 m$^3$ | 19 | 51,510 | 5938.323 | 8541.610 | $7.30 \times 10^7$ |
| Fish eggs | ind./1000 m$^3$ | 0 | 81,596 | 10,418.305 | 14,776.330 | $2.18 \times 10^8$ |
| Fish larvae | ind./1000 m$^3$ | 0 | 3551 | 318.350 | 457.812 | 209,591.670 |

The standardized correlation coefficient (r) ranges between −1 and +1, with a value closer to −1 or +1 indicating a stronger correlation between the two random variables, and a value closer to 0 indicating a weaker correlation. In Table 5, it is evident that a majority of the measurement variables under investigation exhibited a substantial degree of linear relationship, while the remaining variables could not be determined as having a linear relationship (potentially due to a non-linear relationship or lack of correlation).

As correlation analysis only reveals the existence of linear relationships between variables, it does not imply the establishment of a causal relationship. Thus, in order to identify any potential variables and investigate the common variance between each variable, further factor analysis must be conducted.

**Table 5.** Correlation analysis results of measurement variables.

| | Temperature | Salinity | pH | Dissolved oxygen | Transparency | Chlorophyll a | Nitrate | Nitrite | Phosphate | Silicate | Zooplankton Abundance | Phytoplankton Abundance | Crab Larvae Abundance | Shrimp Larvae Abundance | Fish Egg Abundance | Fish Larvae Abundance | Fish Species | Fish Abundance |
|---|---|---|---|---|---|---|---|---|---|---|---|---|---|---|---|---|---|---|
| Temperature | 1.000 | | | | | | | | | | | | | | | | | |
| Salinity | −0.586 ** | 1.000 | | | | | | | | | | | | | | | | |
| pH | 0.444 ** | −0.128 | 1.000 | | | | | | | | | | | | | | | |
| Dissolved oxygen | −0.550 ** | 0.239 ** | −0.014 | 1.000 | | | | | | | | | | | | | | |
| Transparency | 0.074 | 0.088 | −0.074 | −0.325 ** | 1.000 | | | | | | | | | | | | | |
| Chlorophyll a | −0.126 | −0.273 ** | 0.051 | 0.233 ** | −0.395 ** | 1.000 | | | | | | | | | | | | |
| Nitrate | −0.432 ** | 0.268 ** | −0.384** | 0.064 | −0.030 | 0.109 | 1.000 | | | | | | | | | | | |
| Nitrite | −0.307 ** | 0.321 ** | −0.163 * | 0.170 * | −0.122 | 0.065 | 0.285 ** | 1.000 | | | | | | | | | | |
| Phosphate | −0.396 ** | 0.264 ** | −0.280 ** | 0.208 ** | −0.209 ** | 0.199 ** | 0.547 ** | 0.376 ** | 1.000 | | | | | | | | | |
| Silicate | −0.190 ** | −0.101 | −0.248 ** | 0.086 | −0.029 | 0.104 | 0.438 ** | 0.215 ** | 0.351 ** | 1.000 | | | | | | | | |
| Zooplankton abundance | 0.124 | −0.191 ** | −0.236 ** | 0.016 | 0.041 | 0.047 | −0.245 ** | −0.164 * | −0.177 ** | −0.183 ** | 1.000 | | | | | | | |
| Phytoplankton abundance | 0.237 ** | −0.143 * | 0.093 | −0.010 | −0.107 | −0.060 | −0.216 ** | −0.038 | −0.139 * | −0.029 | 0.079 | 1.000 | | | | | | |
| Crab larvae abundance | 0.285 ** | −0.314 ** | 0.044 | −0.126 | −0.184 ** | 0.099 | −0.192 ** | −0.157 * | −0.093 | −0.090 | 0.385 ** | 0.066 | 1.000 | | | | | |
| Shrimp larvae abundance | −0.041 | −0.109 | −0.180 ** | 0.140 * | −0.123 | 0.083 | −0.138 * | −0.132 * | −0.067 | −0.203 ** | 0.634 ** | 0.111 | 0.474 ** | 1.000 | | | | |
| Fish egg abundance | 0.550 ** | −0.346 ** | −0.020 | −0.392 ** | 0.176 ** | −0.076 | −0.172 * | −0.141 * | −0.113 | 0.045 | 0.301 ** | 0.228 ** | 0.375 ** | 0.107 | 1.000 | | | |
| Fish larvae abundance | 0.254 ** | −0.091 | 0.019 | −0.093 | 0.020 | −0.047 | −0.137 * | −0.130 | −0.148 * | −0.213 ** | 0.476 ** | 0.143 * | 0.396 ** | 0.501 ** | 0.347 ** | 1.000 | | |
| Fish species | 0.084 | −0.175 ** | −0.077 | 0.078 | −0.085 | −0.081 | −0.212 ** | −0.061 | −0.172 * | −0.050 | 0.120 | 0.176 ** | 0.133 * | 0.167 * | 0.004 | 0.118 | 1.000 | |
| Fish abundance | 0.049 | 0.006 | 0.173 ** | 0.032 | −0.021 | 0.018 | −0.201 ** | −0.071 | −0.103 | −0.026 | −0.017 | 0.039 | 0.017 | 0.016 | −0.029 | 0.069 | 0.460 ** | 1.000 |

statistical significance *. $p \leq 0.05$; **. $p \leq 0.01$.

*3.2. Factor Analysis of Environmental Variables*

Before conducting factor analysis, it is necessary to determine the suitability of each measurement variable by checking whether the Kaiser–Meyer–Olkin (KMO) value is greater than 0.6 and if the result of Bartlett's sphericity test is significant ($p \leq 0.05$) [13]. This study postulates that environmental factors influence marine life. However, measurable environmental and biological variables (such as nitrate, silicate, phosphate, nitrite, temperature, salinity, pH, dissolved oxygen, transparency, and chlorophyll a, along with the quantity of zooplankton, crab larvae, shrimp larvae, fish eggs, larvae, fish species, and fish abundance) are closely intertwined. Therefore, potential factors were first extracted from all the environmental and biological variables. Among the observed variables related to phytoplankton, only the "number of phytoplankton" was included in the analysis of biological variables, as it can explain the variation in phytoplankton community structure; hence, it was reserved in the SEM but excluded from factor analysis of the biological variables.

3.2.1. First Factor Analysis

The result of the first factor analysis of the environmental variables indicated a KMO value of 0.642. As per Kaiser's suggested criteria, a value between 0.6 and 0.7 was considered to have "normal" applicability. Furthermore, the result of Bartlett's sphericity test was significant ($p < 0.001$), implying that the water quality measurement variables investigated in this study are appropriate for factor analysis.

In this study, the principal component method of extraction was employed to extract factors. Following Kaiser's criterion, only factors with eigenvalues greater than or equal to 1 were retained. The scree plot was also used to observe the slope of the cumulative explanatory power. When the slope is significantly flattened, the extraction process can be stopped. The results indicated that three factors had eigenvalues greater than 1 and could explain 61.075% of the total variation. Factor rotation was then conducted with the determined principal components.

The purpose of factor rotation is to align the data with the assumptions of the statistical model and to convert the data appropriately. It is undertaken by rotating the axis to cover the "maximum space range", which is associated with different factor loadings, thereby amplifying the differences. In other words, it aims to achieve the greatest amount of variation. Through rotation, both positive and negative correlations between each factor (axis) and the variables are strengthened, and thus variables that were initially relevant will maintain a high factor loading, which is conducive to naming and interpreting the factors (latent variable).

In this study, we utilized the Varimax method of orthogonal rotation, which enables each variable to have only one latent factor, yielding a large factor loading while avoiding duplication. Orthogonal rotations preserve a 90-degree angle. The Varimax method facilitates a distinct partition of variables, leading some to achieve high factor loadings and others to have low factor loadings, thereby making the factors easier to interpret.

After rotation, factor 1 (nitrate, silicate, phosphate, nitrite) explained 24.528% of the variance; factor 2 (temperature, dissolved oxygen, salinity) explained 18.505% of the variance; and factor 3 (transparency, chlorophyll a) explained 18.042% of the variance. This result indicated that the pH environmental variable did not demonstrate adequate convergence validity, as none of the three different rotations produced factor loading values above 0.5.

Furthermore, the communality score of the pH value was 0.359, the lowest value among all measurement variables, and did not exceed 0.5. Therefore, since pH did not have convergent validity and yielded low communality, it was removed and a second factor analysis was conducted. Note that when deleting variables, it is essential to remove only one at a time and consider the importance of each variable to the research.

3.2.2. Second Factor Analysis

The second attempt resulted in a KMO value of 0.632, indicating normal applicability. Additionally, Bartlett's sphericity test was significant, with $p < 0.001$. The results showed that after removing pH, the remaining environmental variables were still suitable for factor analysis.

In the second step, the scree plot diagram revealed that three factors had eigenvalues greater than 1, resulting in a cumulative total variation of 65.360%. These factors were further processed in factor rotation.

After the rotation, factor 1 (nitrate, silicate, phosphate, nitrite) could explain 25.860% of the variance; factor 2 (temperature, dissolved oxygen, salinity) could explain 20.347% of the variance; and factor 3 (chlorophyll a, transparency) could explain 19.153% of the variance. It was evident that the "salinity" environmental variables could be observed in both factor composition axes 2 and 3, with both exceeding a factor loading of 0.5. This indicated that salinity lacks discriminant validity. However, none of the other environmental variables from factors 1 to 3 exhibited factor loadings of 0.5 and above simultaneously, implying that these variables possessed discriminant validity. Moreover, none of the environmental variables in factors 1 to 3 had all factor loadings below 0.5, indicating that variables from factor 1 to 3 possessed convergent validity. Lastly, it is essential to examine whether the communality of the environmental variables is greater than or equal to 0.5. As illustrated in Table 6, "nitrite" exhibited a communality of 0.427, which is lower than 0.5, rendering it the smallest of the other environmental variables. Therefore, it was eliminated, and the third factor analysis was executed.

**Table 6.** The communality of environmental variables (pH excluded).

| Environmental Variables | Total Variance Extracted % |
|---|---|
| Temperature | 0.789 |
| Salinity | 0.765 |
| Dissolved oxygen | 0.733 |
| Transparency | 0.606 |
| Chlorophyll a | 0.637 |
| Nitrate | 0.722 |
| Nitrite | 0.427 |
| Phosphate | 0.525 |
| Silicate | 0.678 |

3.2.3. Third Factor Analysis

The third factor analysis yielded a KMO value of 0.579, which is deemed "not a good fit". However, Bartlett's sphericity test was significant ($p < 0.001$), indicating the presence of sufficient correlation among the variables. Nonetheless, given the low effect of extracting common factors, as revealed by KMO, it is not advisable to proceed with further analysis of the remaining environmental variables if nitrite is eliminated.

The sea area of the present study contains interrelationships among various water quality environmental measurement variables. Specifically, the study examined the relationship between phosphate and nitrate, which serve as raw materials for the synthesis of organic matter via the photosynthesis of marine plants, and silicates, which are the primary constituent materials of the phytoplankton cell wall. These interrelationships arise from the interaction between environmental and biological variables. However, given the dynamic nature of marine environments, it was challenging to identify the precise nature of these relationships. Furthermore, deleting any variable may result in interpretational errors. Hence, the researchers chose to exclude only the pH variable. The remaining environmental variables were retained and named based on the outcomes of the second factor analysis, as shown in Tables 6 and 7.

**Table 7.** The component matrix of environmental variables in each factor after rotation (pH excluded).

| Measured Variable | Factor Loading (N = 223) | | |
|---|---|---|---|
| | 1 | 2 | 3 |
| Nitrate | 0.828 | 0.191 | −0.019 |
| Silicate | 0.811 | −0.113 | 0.084 |
| Phosphate | 0.693 | 0.090 | 0.193 |
| Nitrite | 0.584 | 0.290 | −0.053 |
| Temperature | −0.297 | −0.836 | 0.051 |
| Dissolved oxygen | −0.082 | 0.737 | 0.428 |
| Salinity | 0.198 | 0.639 | −0.563 |
| Chlorophyll a | 0.158 | −0.088 | 0.777 |
| Transparency | −0.066 | −0.180 | −0.754 |
| Eigenvalue | 2.237 | 1.831 | 1.724 |
| Variance % | 25.860 | 20.347 | 19.153 |
| Cumulated variance % | 25.860 | 46.207 | 65.360 |

3.2.4. Factor Naming

In the field of factor analysis, each variable possesses a distinct meaning, and the extracted factors themselves hold unique significance. Typically, factors are labeled after variables that display high factor loading, and their collective meaning is synthesized to name the factor. In the present study, water quality samples were obtained from the adjacent sea area of Nanwan Bay, Kenting, Taiwan. Previous research has indicated that the hydrological environment in the nearby waters is intricate, and the occurrence of upwelling in the bay has been established. As a result, the factors were named after Nanwan Bay's ocean environmental variations.

Based on the accumulated findings from various studies conducted in the sea area over the years, and the factor analysis outcomes displayed in Table 6, three component axes were extracted from the component matrix following rotation. These axes are described below.

The first component axis in the present study encompassed nitrate (0.828), silicate (0.811), phosphate (0.693), and nitrite (0.584), which accounted for 25.860% of the variation. The factor loading was positive, indicating a positive correlation among the variables. It is noteworthy that in this study, the majority of the water quality measurement parameters were collected from the water surface. The occurrence of sea surges elevates the nutrient salt from the deep ocean to the surface, leading to a concurrent increase in nutrients. Therefore, this component axis was aptly named "nutrients."

The second component axis in the current study was comprised of temperature (−0.836), dissolved oxygen (0.737), and salinity (0.639), which accounted for 20.347% of the variation. The results showed a negative factor loading for temperature, while dissolved oxygen and salinity exhibited positive factor loading, indicating a negative correlation between temperature and dissolved oxygen, and temperature and salinity, but a positive correlation between dissolved oxygen and salinity.

Taiwan has a subtropical location, and the surface of the seawater is influenced by solar radiation and is typically warmer. Additionally, the evaporation rate exceeds the rainfall rate, leading to an increase in seawater salinity. The waters near Nanwan Bay are impacted by surges, which transport colder water from deep mid-levels to the surface. Deep mid-level waters are characterized by the absence of light, inhibiting photosynthesis, and, therefore, are not saturated with dissolved oxygen. This leads to an overall reduction in both dissolved oxygen and temperature when such waters surface. However, according to other studies, Nanwan Bay is also affected by internal ocean waves. These waves induce intense water accumulation at the seabed, which subsequently elevates the level of

dissolved oxygen at the surface [9]. Given that the location of this study is within an inland bay and considering the highest correlation coefficient between temperature drop in the component axis and factor 2 ($-0.836$), the component axis was named "upwelling current."

The third component axis included chlorophyll a (0.777), transparency ($-0.754$), and salinity ($-0.563$), explaining 19.153% of the variance. Chlorophyll a exhibited a positive factor load, while transparency and salinity had negative factor loads. This indicated a negative correlation between chlorophyll a and both transparency and salinity, while salinity and transparency showed a positive correlation. Since chlorophyll a serves as the primary photosynthetic pigment in marine phytoplankton, an increase in measured chlorophyll a concentration implies the growth of phytoplankton, consequently leading to a reduction in underwater transparency. Moreover, the possibility of decreased salinity was associated with the proliferation of phytoplankton covering the sea surface, which, under severe circumstances, could trigger an algal bloom phenomenon, reducing the intensity of sunlight penetrating below the water surface and resulting in decreased evaporation from the surface water. Hence, based on our findings, the component axis was named "primary production".

### 3.3. Factor Analysis of Biological Variables

After the factor analysis of environmental variables, the next step was to factor analyze biological variables.

### 3.3.1. First Factor Analysis

A KMO value of 0.73 was found, which was considered a "fairly acceptable" fit. Furthermore, Bartlett's sphericity test was significant at $p < 0.001$, indicating the appropriateness of performing factor analysis on the biological variables of interest. The results of factor extraction revealed two eigenvalues greater than 1, which together explained 54.507% of the total variance.

The second step involved performing a factor rotation using the Varimax method of orthogonal rotation. After the rotation, factor 1 (shrimp, number of zooplankton, larvae, crabs) explained 36.064% of the variance and factor 2 (number of fish, number of fish species, fish eggs) explained 18.444% of the variation. Based on the findings, it appears that the biological variables related to the "fish eggs" exhibited factor loadings of less than 0.5 in both component axes 1 and 2, indicating a lack of convergent validity. To determine the adequacy of the remaining variables, the communality values of factors 1 and 2 were examined, with a threshold of 0.5 or greater. The communality value for "fish eggs" was only 0.162, the lowest among all the biological variables. Therefore, "fish eggs" was removed for the second round of factor analysis.

### 3.3.2. Second Factor Analysis

The results of the second factor analysis presented a KMO value of 0.749, indicating a "fair" fit. Moreover, Bartlett's sphericity test was significant ($p < 0.001$), suggesting that the biological variables were appropriate for factor analysis after the removal of the "fish eggs" variable.

After the factor extraction, only two components were reserved for the factor rotation process. Rotation factor 1 (shrimp, number of zooplankton, juvenile larvae, crabs) explained 42.180% variation and factor 2 (number of fish, number of fish species) explained 20.377% of the variation, while none of the factor loadings in factors 1 and 2 were greater than 0.5 simultaneously, which suggested that factors 1 and 2 had discriminant validity. Hence, the results from the second factor analysis were preserved for further analysis, as shown in Table 8.

**Table 8.** The component matrix of biological variables in each factor after rotation (fish eggs excluded).

| Measured Variable | Factor Loading (N = 223) | |
|---|---|---|
| | 1 | 2 |
| Shrimp larvae abundance | 0.886 | 0.073 |
| Zooplankton abundance | 0.855 | −0.004 |
| Fish larvae abundance | 0.773 | 0.174 |
| Crab larvae abundance | 0.634 | −0.053 |
| Fish abundance | −0.039 | 0.779 |
| Fish species | 0.114 | 0.759 |
| Eigenvalue | 2.531 | 1.223 |
| Variance % | 42.180 | 20.377 |
| Cumulated variance % | 42.180 | 62.557 |

### 3.3.3. Factor Naming

Based on the results of the previous studies in the sea area and the analysis of the factor extraction in Table 8, the two factors were named as described below.

The first principal component axis was comprised of four biological variables, namely shrimp larvae (0.886), the number of zooplankton (0.855), fish larvae (0.773), and juvenile crabs (0.634), which collectively accounted for 42.180% of the total variation. The factor loading for each variable was positive, implying a positive correlation between them. Zooplankton, in particular, was widely distributed and had a larger number of species, including copepods. Fish larvae and juvenile crabs were also ecologically significant in terms of fishery resources. Previous studies have shown that the intersection of Kuroshio and upwelling currents support diverse flora and fauna [25,26]. The number of juvenile shrimp (0.886) and zooplankton (0.855) in the first component axis exhibited a higher correlation coefficient with factor 1, indicating that the sea during the previous sampling period had a higher abundance of zooplankton, especially crustaceans, which are the main food source for fish larvae. Consequently, this component axis was termed "zooplankton cluster".

The second component axis was composed of two variables, namely the number of fish (0.779) and the number of fish species (0.759), which explained 42.180% of the variation. The factor loadings of both variables were positive, suggesting a positive correlation between them. The phenomenon of fish migration in groups during the foraging season and the formation of fisheries in areas with abundant zooplankton can lead to a higher number of fish [27]. Thus, the name "fish cluster" was given to this component axis.

### 3.4. Structural Equation

Given the different units of measurements among the variables in the proposed structure, factor analysis was utilized to extract potential factors, namely nutrients, upwelling current, primary productivity, zooplankton cluster, and fish cluster, to investigate the interplay between water quality and plankton assemblage, as well as plankton clustering and fish clustering. To test the hypothesis model, we employed the sampling data, assuming a normal distribution. The estimation model assumed that the measured variables of the latent factors were consistent with those presented in Tables 6 and 8. Notably, the sign of coefficients of the latent factor "upwelling current" differed from the factor loading in Table 6, where the negative values were denoted by the [-] symbol in the model.

### 3.4.1. Water Quality Environmental Factors and Phytoplankton Cluster

Figure 3 shows the structural pattern between water quality environments and phytoplankton cluster. The RMSEA (0.113) fell within the "bad fit" range, indicating that the setting of the study model was not effectively matched with the sampling data. Addition-

ally, the rest of the indicators did not meet the reference criteria. Hence, the overall model did not pass the test.

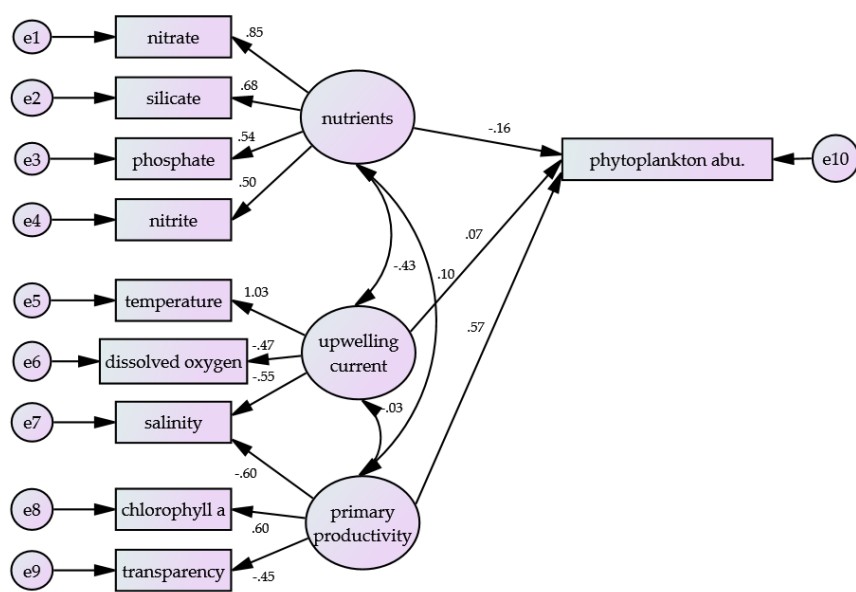

**Figure 3.** SEM of environmental variables and the phytoplankton cluster.

3.4.2. Water Quality Environmental Factors and Zooplankton Cluster

Figure 4 shows the structural pattern between water quality environments and the zooplankton cluster. The RMSEA (0.085) value obtained from the model fit analysis (0.085) fell within the range of "moderately fit", albeit falling short of the optimal reference value of less than 0.05. However, the obtained value was still considered acceptable, suggesting that the conceptual model proposed was in line with the empirical data obtained. Meanwhile, the non-normed fit index (NNFI) value (0.842) did not meet the reference criteria, which was used to assess the degree of association between the research model and the observed variables, and to identify areas for model improvement. Hence, adjustment is needed for covariate relationships.

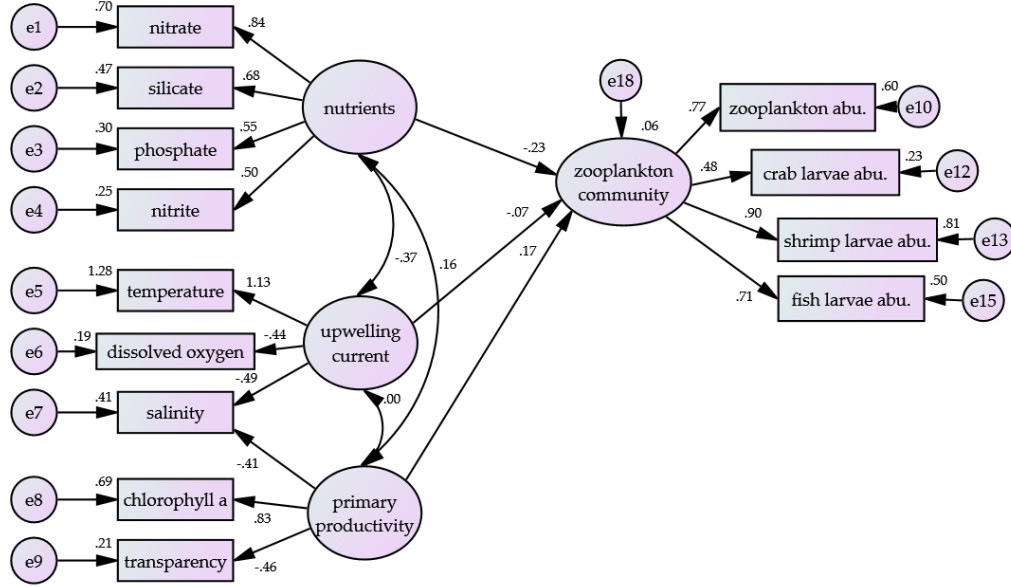

**Figure 4.** SEM of environmental variables and the zooplankton cluster.

### 3.4.3. Water Quality Environmental Factors and Plankton Cluster

Figure 5 presents the structural pattern between water quality environments and both phytoplankton and zooplankton clusters. The RMSEA index displayed a value of 0.097, which indicates a "moderate" fit; however, it was close to the threshold of a poor fit. This suggested that the model and the sampling data had only a low probability of effectively explaining the results. Furthermore, the NNFI value (0.787) did not meet the reference criteria. The GFI value (0.892) also failed to meet the reference criteria. GFI is primarily used to test the proportion between the variance of the explainable observed variables before model adjustment and covariance. On the other hand, AGFI (0.831) met the criteria, suggesting the need for more observational data to enhance the interpretation of the observed and potential variables in the model.

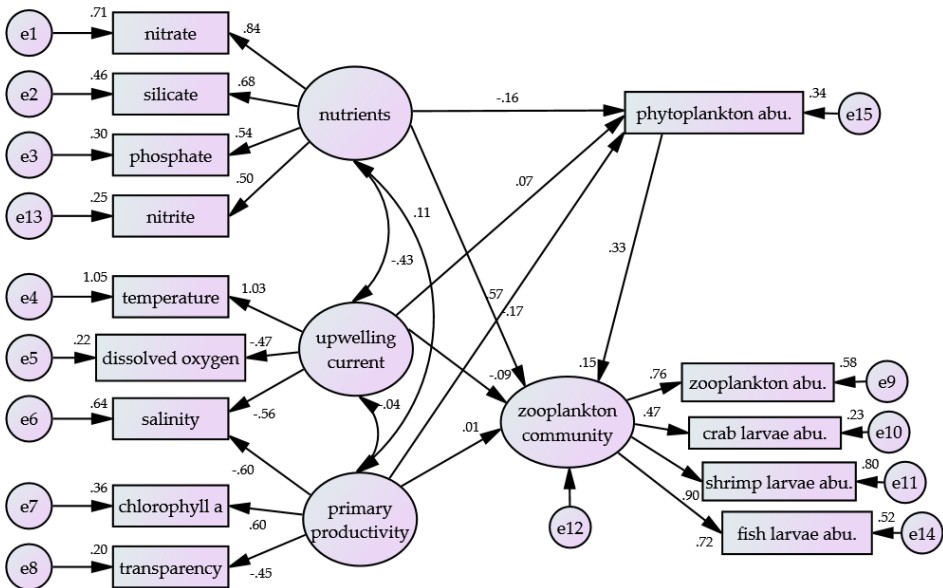

**Figure 5.** SEM of environmental variables and plankton clusters.

### 3.4.4. Water Quality Environmental Factors and Marine Life Cluster

The structure that included all three clusters was also considered. However, due to the high correlation between chlorophyll a and the phytoplankton cluster, we reserved chlorophyll a to represent the primary productivity and dropped the phytoplankton cluster.

Figure 6 shows the comprehensive structure pattern between water quality environments and both zooplankton and fish clusters. Of all the indices utilized, only NNFI (0.840) fell short of meeting the reference value, therefore, requiring the further refinement of the model. The remaining indices successfully passed the test, with RMSEA (0.074) reaching a level of good fit. This suggested that the model has the potential to effectively explicate marine ecological phenomena to a significant extent.

This model which examined the relationship between environmental factors and marine life clusters failed to meet the NNFI criteria. This outcome may be attributed to several factors, including the nature of the sample itself; environmental changes, such as seasonal and weekly–daily fluctuations; and the accuracy and stability of the measuring instrument. These factors may result in a higher probability of standard errors (non-normal distribution) in the measurement variables. Furthermore, various potential environmental factors in Nanway Bay, such as sea tides and internal wave phenomena, were not included in the statistical analysis. Consequently, the proposed model may be limited in its ability to explain ecological phenomena solely within the sampled waters and may not be applicable to other water bodies.

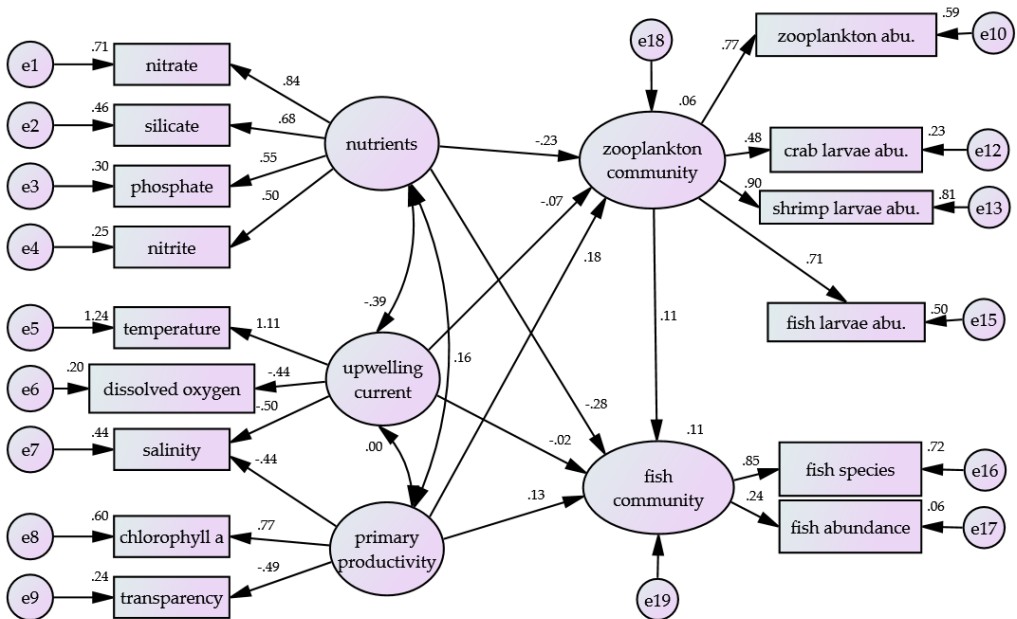

**Figure 6.** SEM of environmental variables and marine life.

### 3.5. Model Modification

Continuing with the results of the model verification, the next step involved model modification. Due to the covariant relationship between the observed variables in the model, parameters in the modification index (MI) provided in the Amos Graphics software could be used to modify the model. The main objectives of model modification were to improve its simplicity, model fit, explanatory power, and reduce measurement error and structural residuals. However, there was a risk of losing the characteristic of verification and converting the model into an exploratory tool. In the context of the measurement model, one way to modify the model is to allow correlation between measurement variables when supported by theory or the literature.

The objective of this study was to explore the correlation between the variables presented in Figures 5 and 6. During the model verification phase, the root mean square error of approximation (RMSEA) of the models, shown in Figures 4–6, were all in the moderate fit range (0.08 to 0.10). Notably, the model depicted in Figure 6 achieved a better fit range (0.05 to 0.08). As the correlation of the variables in the model of Figure 4 was included in the model of Figure 5, only the models in Figures 5 and 6 were considered for revision.

The model in Figure 5 was modified based on the MI value provided in the Amos report by establishing the correlation between the residuals of measured variables. Specifically, the correlation between measured variable residuals was increased to reduce the Chi-square value, following the principle of modifying one parameter at a time. The revised model is shown in Figure 7. After the revision, the NNFI (0.939) was in accordance with the reference criteria, and the rest of the indices also provided validation for the model; RMSEA (0.052), especially, reached the well-fit range, indicating that the model and observed data achieved the desired fit (Table 9).

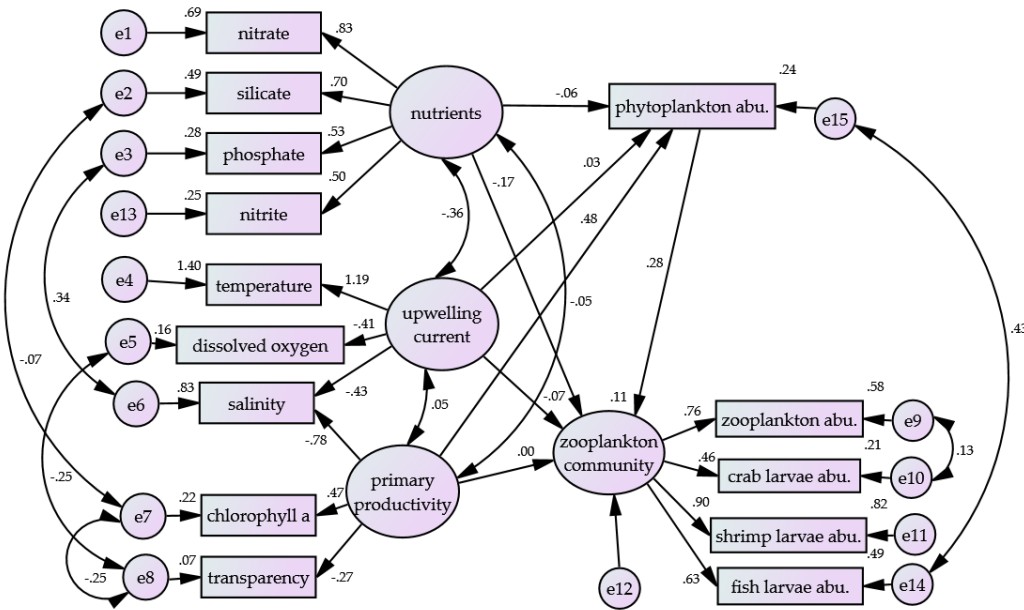

**Figure 7.** Modified SEM of environmental variables and plankton clusters.

**Table 9.** The model evaluation results of the modified SEM of environmental variables and the plankton clusters.

| Key Metrics | Reference Guidelines | Model Validation | Test Results |
|---|---|---|---|
| $\chi^2$ | The lower the better | 97.663 | |
| $\chi^2/\text{df}$ | <5(<3 better fit) | 1.601 | Compliant |
| GFI | >0.9 | 0.941 | Compliant |
| AGFI | >0.9(>0.8 acceptable fit) | 0.898 | Compliant |
| RMSEA | <0.05, good fit | 0.052 | Reasonable fit |
| | 0.05~0.08, reasonable fit | | |
| | 0.08~0.10, medium fit | | |
| | >0.10, poor fit | | |
| SRMR | ≤0.08 | 0.0634 | Compliant |
| NNFI | >0.9 | 0.939 | Compliant |
| PNFI | ≥0.5 | 0.604 | Compliant |
| PGFI | ≥0.5 | 0.547 | Compliant |
| GN | ≥200 | 223 | Compliant |

Figure 8 is the revision of Figure 6, and the results are shown in Table 10. Notice that the normalized fit index (NNFI) attained a value of 0.912, satisfying the established reference criteria. Furthermore, the other indices also provided validation of the model, with particular emphasis on RMSEA, which fell within the well-fit range at 0.055. This indicated that the model and the observed data achieved the desired level of fit.

Upon completion of the model revision, a subsequent path analysis and the effect between variables were conducted to verify the assumptions made in this study.

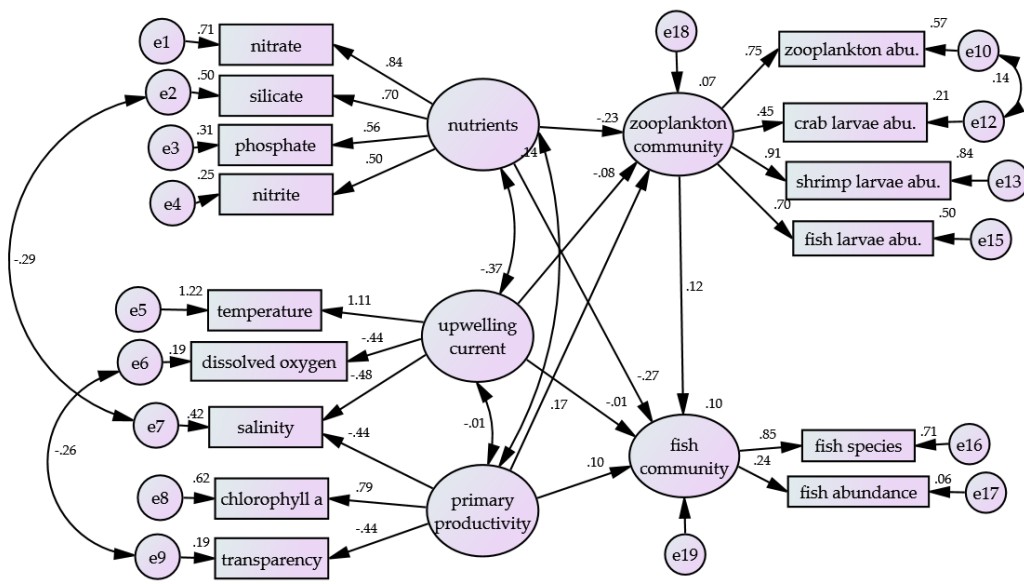

**Figure 8.** Modified SEM of environmental variables and marine life.

**Table 10.** The model evaluation results of the modified SEM of environmental variables and marine life.

| Key Metrics | Reference Guidelines | Model Validation | Test Result |
|---|---|---|---|
| $\chi^2$ | The lower the better | 127.053 | |
| $\chi^2/df$ | <5(<3 better fit) | 1.672 | Compliant |
| GFI | >0.9 | 0.931 | Compliant |
| AGFI | >0.9(>0.8 acceptable fit) | 0.891 | Compliant |
| RMSEA | <0.05, good fit<br>0.05~0.08, reasonable fit<br>0.08~0.10, medium fit<br>>0.10, poor fit | 0.055 | Reasonable fit |
| SRMR | ≤0.08 | 0.0621 | Compliant |
| NNFI | >0.9 | 0.912 | Compliant |
| PNFI | ≥0.5 | 0.623 | Compliant |
| PGFI | ≥0.5 | 0.589 | Compliant |
| GN | ≥200 | 223 | Compliant |

*3.6. Path Analysis*

In addition to evaluating the overall fitness of the model modification and the intrinsic quality of the test, further examination was required to comprehend the linear association between the latent variables. This can be achieved through the observed direct effects and indirect effects to determine the direct and indirect impacts, as well as overall impacts (direct and indirect effects) among the latent variables.

The path relationships between facets were estimated using the structural equation model. Standardized coefficients were used to determine the relationship between the latent variables in the model, as depicted in Figures 7 and 8. In Figure 9, the path effects of "nutrient on zooplankton clustering," "primary productivity on phytoplankton clustering", and "phytoplankton clustering on zooplankton clustering" were found to be statistically significant. Similarly, in Figure 10, the path effects of "nutrient on zooplankton clustering" and "primary productivity on zooplankton clustering" were also significant, indicating

that both models possessed considerable predictive capabilities for assessing direct and indirect effects (enhancement or offset) on environmental and biological factors.

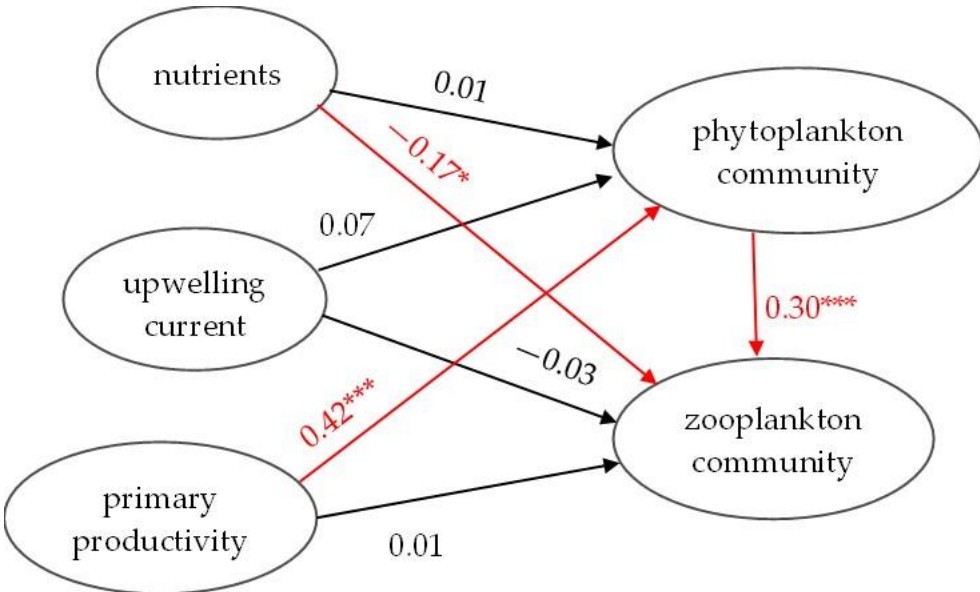

statistical significance *. $p \leq 0.05$; *** $p \leq 0.001$

**Figure 9.** The path analysis of environmental variables and plankton clusters.

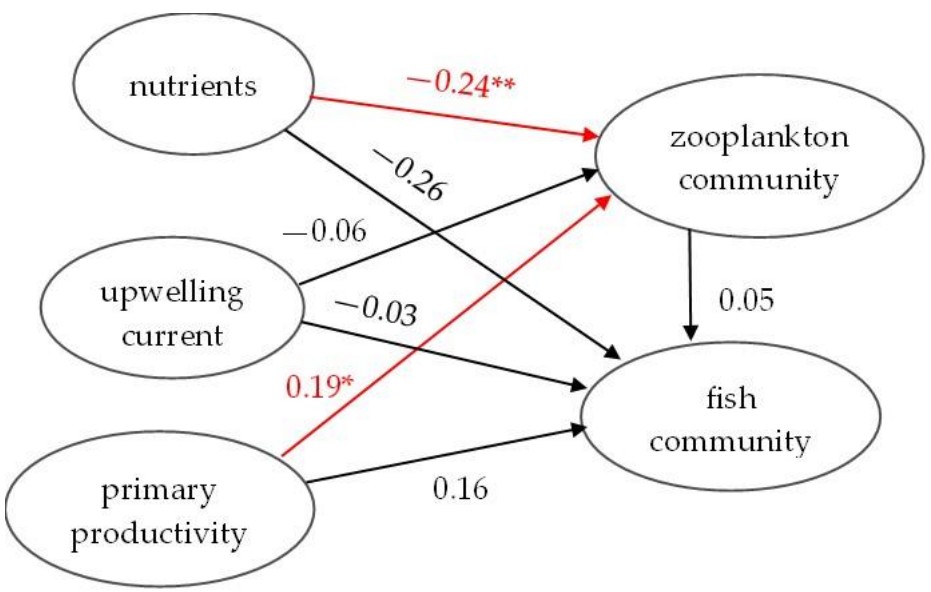

statistical significance *. $p \leq 0.05$; **. $p \leq 0.01$

**Figure 10.** The path analysis of environmental variables and marine life.

The path analysis provides empirical evidence of the direct and indirect effects. The direct effects of nutrients on the zooplankton cluster, primary productivity on the phytoplankton cluster, and phytoplankton cluster on the zooplankton cluster were found to be statistically significant (H2, H5, and H7, respectively). Additionally, the direct effects of nutrients on the zooplankton cluster and primary productivity on the zooplankton cluster were also statistically significant (H8 and H12, respectively). Among the significant

direct effects, the effect of primary productivity on the phytoplankton cluster (H5) was the strongest (0.421).

In addition to the direct effects, the study also examined the indirect effects of the predictor variables on the zooplankton cluster. The results in Tables 11 and 12 indicated that, except for the path of primary productivity on the zooplankton cluster, which had a rather higher coefficient of 0.122, the remaining paths had lower coefficients. Therefore, the direct effects were found to be more significant than the indirect effects.

**Table 11.** The effects of environmental factors and plankton cluster.

|  |  | Phytoplankton | Zooplankton |
|---|---|---|---|
| Nutrients | Direct effect | 0.009 (H1) | −0.172 * (H2) |
|  | Indirect effect | - | 0.003 |
|  | Total effect | 0.009 | −0.169 |
| Upwelling Current | Direct effect | 0.072 (H3) | −0.033 (H4) |
|  | Indirect effect | - | 0.0231 |
|  | Total effect | 0.072 | −0.012 |
| Primary Productivity | Direct effect | 0.421 *** (H5) | 0.012 (H6) |
|  | Indirect effect | - | 0.122 |
|  | Total effect | 0.421 | 0.135 |
| Phytoplankton | Direct effect | - | 0.290 *** (H7) |
|  | Indirect effect | - | - |
|  | Total effect | - | 0.290 |

\* $p < 0.05$; \*\*\* $p < 0.001$; different superscripts indicate significant difference.

**Table 12.** The effects of environmental factors and marine life.

|  |  | Zooplankton | Fish Species |
|---|---|---|---|
| Nutrients | Direct effect | −0.239 ** (H8) | −0.265 (H9) |
|  | Indirect effect | - | −0.011 |
|  | Total effect | −0.239 | −0.276 |
| Upwelling Current | Direct effect | −0.056 (H10) | −0.034 (H11) |
|  | Indirect effect | - | −0.003 |
|  | Total effect | −0.056 | −0.036 |
| Primary Productivity | Direct effect | 0.192 * (H12) | 0.159 (H13) |
|  | Indirect effect | - | 0.006 |
|  | Total effect | 0.192 | 0.168 |
| Zooplankton | Direct effect | - | 0.048 (H14) |
|  | Indirect effect | - | - |
|  | Total effect | - | 0.048 |

\* $p < 0.05$; \*\* $p < 0.01$; different superscripts indicate significant difference.

Notably, despite being presented as a latent variable in the SEM model, the upwelling current did not display any direct or indirect effects on plankton and fish assemblages in Nanwan Bay. It is worth mentioning that areas characterized by upwelling currents are commonly known to provide ideal fishing grounds due to the abundant nutrient supply for planktonic communities [28]. Therefore, it is plausible that the upwelling current may indirectly influence other environmental factors, such as nutrient availability and

primary productivity, which in turn have an impact on the distribution and abundance of phytoplankton, zooplankton, and fish.

Overall, the study suggested that the marine environment was subject to various factors that may influence the relationships between nutrient salt, primary productivity, phytoplankton cluster, and zooplankton cluster. This may explain why the indirect effects were not significant in this study. It is also possible that there were other intermediary variables or relationships that were not included in the structural statistics, or that the data themselves had a high degree of variation.

## 4. Conclusions

Our investigation aimed to unravel the intricate connections between environmental factors and marine life in Nanwan Bay. By employing structural equation modeling (SEM), we uncovered significant findings that illuminated the complex dynamics within the bay's marine ecosystem. These insights provide valuable contributions to our understanding of the relationships and dependencies among environmental factors and the organisms inhabiting Nanwan Bay.

The first notable finding emerged from the modified models developed in this study, namely "modification of environmental factors and plankton clusters" and "modification of environmental factors and marine life clusters". These models identified "primary productivity" and "nutrient" as the main environmental change factors, with a considerable degree of impact on the abundance and distribution of plankton clusters. Notably, the analysis revealed that primary productivity exhibited the highest direct effect on plankton clusters, emphasizing its pivotal role in shaping plankton communities.

Building upon the findings from the modified models, the second finding explored the relationship between the upwelling current and the phytoplankton cluster, zooplankton cluster, and fish cluster. Surprisingly, the path analysis indicated that the upwelling current did not have a statistically significant effect on these variables. This suggests that the direct impact of the upwelling current on the studied marine organisms might be limited, despite playing a role in the SEM models.

The proposed SEM offers valuable insights into the intricate relationships between environmental factors and marine organisms, particularly in the context of water quality, plankton communities, and fish populations. The findings highlight the significant influence of primary productivity and nutrients while also providing a nuanced understanding of the role of the upwelling current in the study area.

**Author Contributions:** All authors contributed to the designed research, analyzed the data, and wrote up the paper. All authors have read and agreed to the published version of the manuscript.

**Funding:** This research received no external funding.

**Institutional Review Board Statement:** Not applicable.

**Informed Consent Statement:** Not applicable.

**Data Availability Statement:** The data utilized in this study is proprietary to Taipower, and as such, will not be available for public dissemination.

**Acknowledgments:** This research was supported by Taipower, Taiwan (grant no. 061090002101). We extend our heartfelt gratitude to the reviewers and editors whose invaluable time and rigorous efforts significantly contributed to the refinement and ultimate realization of this work.

**Conflicts of Interest:** The authors declare no conflict of interest.

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
