# Peer review of "Structural Equation Modeling of the Marine Ecological System in Nanwan Bay Using SPSS Amos"

_sustainability, doi:10.3390/su151411435_

Round 1

Reviewer 1 Report

Abstract

 The abstract appears to be a deviation from what the authors studied. There are too many inconsistent and unclear statements that must be improved upon. Please, see the attached document for detailed comments.

Introduction

The introduction was poorly written. I found it difficult to understand the argument of the authors. What is the study all about? There are too many inconsistent and confusing statements. The arguments were not flowing, hence the hard to understand arguments.

I would advise the authors to revisit this section and tell us taking one point at a time and built on it before moving to the next point.

Materials and methods   

This section was general well written but can be improved upon as Figure 2 was a little bit not clear.     

Results and discussion      

The results was well crafted but poorly presented. Results are presented in reported speech but the authors were presenting in present terms and present continuous. The study has been concluded and you are to report to trying to report. Please revisit the results section. Also, I would advise you give the manuscript to a first English speaker to help with the grammar.

To the best of my knowledge the authors did not discuss their findings. There appears to be areas were the author attempted discussing their findings, but it was not clear whether they were discussing or just presenting their results. I would advise the authors should discuss their findings with relevant literature.

Please separate the results from the discussion for clarity.

The authors should consult a first English speaker to help them with the grammar. Otherwise, they can use the edit English package by MDPI to improve on the manuscript grammar. 

Author Response

Dear Reviewer,

Thank you for the detailed review and your valuable suggestions. We greatly appreciate your feedback, and have carefully considered each of your comments.

We would like to express our apologies for any editorial errors that may have been present in the initial submission. We have conducted a thorough proofreading of the manuscript to address any grammar mistakes and semantic errors.

Please find attached a point-by-point explanation of the revisions we have made based on your feedback.

Thank you once again for your valuable time and suggestions.

Best regards,

Dr. Jung-Fu Huang

Reviewer 2 Report

The rationale behind the project is interesting and somehow innovative, provide study used Structural Equation Models (SEM), which integrates Factor Analysis and Path Analysis, to analyze the data and incorporates multiple environmental and biological factors into one model. The goal is to explore the extent of the impact of natural and anthropogenic interference on the accumulation of marine organisms through a appropriate model and quantitative analysis. In general, I do find this manuscript being overall very well written; the authors might consider a final proofing paying special attention some editorial errors but I do not consider it is a reviewer’s duty to care for these.The revisions should basically focus on putting the research within the context of a scientific hypothesis to be tested, rather than a write-up of a descriptive report and thus improve the conclusion.

English could be improved with minor editing

Author Response

Dear Reviewer,

Thank you for your positive feedback, and we appreciate your recognition of the innovative use of Structural Equation Models (SEM) to explore the impact of environmental factors on marine ecosystems.

We acknowledge your suggestion to further improve the conclusion section and put the research within the context of the scientific hypothesis to be tested. In the revision, you will notice that we have clearly stated the hypothesis in the introduction section, supported by relevant references. Furthermore, in the conclusion section, we have highlighted how our findings align with the hypothesis.

Regarding editorial errors, we have conducted a thorough proofreading and have corrected all grammar mistakes and semantic errors.

 Please see the attachment for a detailed point by point explanation of the revision.

Thank you once again for your valuable feedback and suggestions.

Best regards,

Dr. Jung-Fu Huang

Reviewer 3 Report

Congratulation your long-term, interdisciplinary case study of high value., including application larvae of shrimps and fishes,; - yuvenile studes were recommended as especially sensitive to environmental factors at the team international monograph "Environmental Quality and Safty"eds.Coulrston, Korte, 1976,  later supported by OECD experts etc. I propose to supplement your study by laboratory experiments following your results;  for testing           the influence of reported by your team changes of water Temperature focused on the influence of ; dissolved Carbon Dioxide in water, available to Phytoplankton and co-influence of changes of Transparency with exposition phytoplankton to Solar Radiation (incl. different wavelenghts) and chlorophyl a  concentration, supplemented  with evaluation availability of Magnesium and correlation with Primary Production of biomas  and homeostasis of  invesitgated marine ecosystems., as model one...

Author Response

Dear Reviewer, 

Thank you for your kind words and suggestions. I appreciate your recommendation to conduct supplementary experiments, and we believe this can be a potential topic for further study. Investigating the influences of water temperature, dissolved carbon dioxide levels, transparency, solar radiation exposure, chlorophyll-a concentration, magnesium availability, and their correlations with primary production and ecosystem homeostasis could indeed be an interesting area of research to explore in the future.

Please see the attachment for point by point explanation of the revision.

Thank you for your time.

Best,

Dr. Jung-Fu Huang

Round 2

Reviewer 1 Report

Dear authors,

Thank you for resubmitting the revised version of your manuscript for reconsideration. It is clear that a substantial efforts have be put in place in addressing most of my concerns.

However, I would like you to look at the manuscript one more time to improve the grammar, may be give it to a first English speaker to help with the grammar.

I also noticed you attached response note to reviewers, but your responses were embedded in the manuscript, which is okay by me, but some of the questions I raised that needed explanation were not addressed, though some changes were made in that regards. I would like you to respond to those questions one after the other for clarity.

Thanks.

The quality of English has been improved in this revised version of the manuscript.

Author Response

Dear Reviewer,

Thank you once again for your time and effort in reviewing our work. In response to the consistent grammar errors, we have sought the English editing service provided by MDPI for this revised version. Therefore, the following point-by-point explanation addresses only the non-grammatical or semantic questions raised in the first round of review.

For detailed responses, please refer to the attached document.

Best regards,

Jung-Fu Huang
